# Better accuracy of robotic-assisted total knee arthroplasty compared to conventional technique in patients with failed high tibial osteotomy

Ji-Hoon Baek[1], Su Chan Lee[1], Dong Nyoung Lee[1], Juneyoung Heo[2], Taehyeon Kim[1], Hye Sun Ahn[1], Chang Hyun Nam [ID][1] *

1 Joint & Arthritis Research, Department of Orthopaedic Surgery, Himchan Hospital, Seoul, Republic of Korea, 2 Joint & Arthritis Research, Department of Neurosurgery, Himchan Hospital, Seoul, Republic of Korea

* himchanhospital@gmail.com

**Data Availability Statement:** All relevant data are within the manuscript.

**Funding:** The author(s) received no specific funding for this work.

## Abstract

### Purpose

This study aimed to compare the clinical outcomes, mechanical axis, component positioning, leg length discrepancy (LLD), and polyethylene liner thickness between robotic-assisted total knee arthroplasty (TKA) and conventional TKA in patients with failed high tibial osteotomy (HTO).

### Methods

A total of 30 patients (30 knees) with failed HTO who underwent TKA using a robot-assisted system between June 2020 and December 2023 were included in this study (robotic group). Additionally, 60 patients (60 knees) with failed HTO who underwent conventional TKA were included as controls (conventional group). Propensity score matching was performed using a 2:1 ratio between the matched participants. The mean follow-up period was 2.1 years in the robotic group and 2.2 years in the conventional group. Clinical evaluations were performed using the Knee Society Score (KSS) rating system. Mechanical axis, component coronal and sagittal positioning, and LLD were evaluated using postoperative radiographs. The thickness of the polyethylene liner was also determined. The mean error values and outliers were calculated and compared between the two groups to determine the accuracy of the mechanical axis, postoperative component positioning, and LLD.

### Results

The postoperative KSSs in the robotic and conventional groups were not statistically different. The robotic group achieved better accuracy than the conventional group in terms of postoperative mean mechanical axis (1.7° vs. 2.4°, p < 0.05), femur coronal inclination (90.0° vs. 91.6°, p < 0.05), tibial coronal inclination (90.3° vs. 91.3°, p < 0.05), tibial sagittal

**Competing interests:** The authors have declared that no competing interests exist.

inclination (90.5˚ vs. 91.4˚, p < 0.05), and LLD (2.2 vs. 7.0 mm, p < 0.05). A significant difference in polyethylene liner thickness was observed between the two groups (p < 0.05).

## Conclusions

Robotic-assisted TKA showed improved mechanical axis, higher accuracy of component positioning and polyethylene liner thickness, and reduced LLD compared with those of conventional TKA in patients with failed HTO. Further studies with a larger sample size and long-term follow-up are warranted to ascertain whether the accuracy of robotic-assisted TKA can translate into better clinical outcomes and patient satisfaction.

## Introduction

High tibial osteotomy (HTO) is an effective surgical treatment for moderate osteoarthritis of the medial compartment of the knee in young active patients with varus deformity [1, 2]. However, because most patients who undergo this procedure are relatively young, clinical improvement may fade over time. Failure of HTO is defined as the need for conversion to total knee arthroplasty (TKA) since HTO is offered as an option to delay the need for knee arthroplasty. The 15-year survival rate of HTO is 30%–90% [3, 4]. Therefore, patients with failed HTO require subsequent TKA, and conversion to TKA due to failure, such as progression of degenerative osteoarthritis and loss of the correction angle, is relatively common [5, 6].

Conversion TKA after HTO may be technically more demanding than primary TKA because of alterations in the mechanical axis and native tibial anatomy, complexity of the approach, and residual ligament imbalance [7]. Conversion TKA after failed HTO is reported to be a challenging procedure with a high risk of revision and complications [8–10]. Successful conversion TKA requires accurate implant positioning and mechanical alignment.

Robotic-assisted TKA was developed to achieve more proper implant positioning and mechanical alignment than conventional TKA. The robotic-assisted TKA system (Mako) was introduced to our hospital in 2020. Mako (Stryker, Kalamazoo, MI, USA) is a leading semi-active robotic system and is currently the most commonly used surgical robot for joint operations worldwide [11]. Mako uses preoperative lower extremity computed tomography (CT) images to preplan the size and position of the implant. During surgery, actual knee bone information and preoperative CT data are used to determine the implant size, position, and extent of bone resection. The bone cutting is performed with a cutting saw mounted on the robotic arm. This robotic arm provides haptic feedback and stops the saw when it starts to exceed a preset range while cutting the bone, preventing damage to soft tissue. Many studies have shown that the accuracy of implant positioning is improved using robotic-assisted systems, and the risk of mechanical axis outliers is decreased compared with those in conventional surgery [12–14]. However, few studies have specifically addressed the clinical and radiological results of robotic-assisted TKA in patients with failed HTO. Predicting implant position, joint line restoration, and leg length discrepancy (LLD) following TKA can provide valuable insights for more effective management of patient expectations and implant longevity in patients with failed HTO.

Therefore, this retrospective case–control study aimed to compare the clinical outcome, mechanical axis, component positioning, LLD, and polyethylene liner thickness between robotic-assisted TKA and conventional TKA in patients with failed HTO. We hypothesized that robotic-assisted TKA would result in better mechanical axis alignment, higher accuracy of

component positioning and polyethylene liner thickness, and lower LLD than those of conventional TKA in patients with failed HTO.

## Materials and methods

### Ethics statement

The design and protocol of this retrospective study were approved by the Institutional Review Board (IRB) of our hospital (IRB number: 116655-01-202401-01). The requirement for informed consent was waived because of the retrospective nature of this study. Data were accessed for the presented analyses on June 30, 2024.

### Study population and data collection

The robotic-assisted TKA system was introduced to our hospital in June 2020. A consecutive series of 40 TKAs was performed in 35 patients with failed HTO between June 2020 and December 2023 at our hospital using the MAKO robotic system. Of the 35 patients, 5 who underwent bilateral TKA were excluded from the study. The final study cohort comprised 30 patients (30 knees, 23 females and 7 males) (robotic group). A total of 60 age-, sex-, and body mass index (BMI)-matched patients with failed HTO who underwent conventional TKA between June 2020 and December 2023 at our hospital were included as controls. Propensity score matching was performed using a 2:1 matching ratio. In total, 60 patients (46 females and 14 males) were included in the conventional group. During this study, we provided sufficient and appropriate information about the respective potential risks, benefits, and specific advantages of robotic and conventional surgeries [15]. Demographic data, including age, sex, BMI, time from HTO to TKA, initial diagnosis, and preoperative Knee Society Score (KSS) [16], were obtained from medical records (Table 1). The mean follow-up period was 2.1 years (range, 0.5–4 years) in the robotic group and 2.2 years (range, 0.5–4 years) in the conventional group.

### Surgical procedure

All surgeries were performed at our hospital by two experienced surgeons (JH Baek and CH Nam) for the robotic and conventional groups. A posterior-stabilized Triathlon total knee prosthesis (Stryker Orthopaedics) was used in all patients. The preoperative goals were neutral alignment and a polyethylene liner thickness of 9 mm in both groups. TKAs were performed using a standard medial parapatellar approach, and the patella was everted laterally. The patella

**Table 1. Patient demographics.**

|  | Robotic group | Conventional group | *p*-value |
|---|---|---|---|
| Cases (patients) | 30 (30) | 60 (60) |  |
| Age (years) | 65.5 ± 65.3 | 64.8 ± 5.0 | 0.541 |
| Sex (female:male) | 23:7 | 46:14 | 1.000 |
| BMI (kg/m$^2$) | 25.8 ± 2.3 | 25.7 ± 3.3 | 0.836 |
| Time from HTO to TKA (year) | 7.4 ± 3.8 | 6.8 ± 3.4 | 0.433 |
| Diagnosis, *n* (%) |  |  |  |
| Osteoarthritis | 30 (100) | 60 (100) | 1.000 |
| Preoperative |  |  |  |
| Knee Society knee score | 36.2 ± 13.9 | 35.4 ± 15.6 | 0.824 |
| Function score | 38.7 ± 9.7 | 38.2 ± 9.7 | 0.818 |

*BMI*, body mass index; *HTO*, High tibial osteotomy; *TKA*, Total knee arthroplasty.

was not replaced. Only the osteophyte was removed in all patients. With the patient's knee joint flexed at approximately 130˚, the anterior cruciate ligament was excised from the femoral notch and tibial attachment area, and the posterior cruciate ligament was removed from the notch.

For robotic-assisted TKA, preoperative CT scans were performed from the hip joint through the knee joint to the ankle joint. The data were input into the robotic software to determine the optimal implant size and position for the patient's knee. A checkpoint and tracking array were installed on the distal femur and tibial shaft [17]. Correction of robotic landmarks, bone registration, and probe verification were performed to confirm the actual positions of the femoral and tibial bones and limb alignment. All ligaments were balanced to ensure adequate tension at the gap between maximum extension and 90˚ flexion of the knee joint. The surgeon confirmed the appropriate implant orientation and position based on robotic verification and balancing of the ligament gap. These parameters were defined and stored in the robotic system before surgery. Resection of the distal femur, posterior chamfer, anterior cortex, anterior chamfer, posterior condyle, and proximal tibia was performed using a robotic arm saw within the virtual boundaries defined by the robot to protect the soft tissues. After femoral box cutting, the femoral and tibial trial implants and the thickness of the liner were assessed. The femoral and tibial implants were implanted using bone cement, and a polyethylene liner was fixed between them.

In the conventional TKA procedure, an intramedullary canal was created by drilling a hole approximately 1 cm anterior to the center of the intercondylar notch. A femoral alignment guide was inserted through the intramedullary hole. Once the desired angle (valgus 5˚) was achieved and the instrument was placed in the appropriate notch, distal femoral resection was performed. An assembly flush was placed on the resected distal femur, and the appropriate femur size and rotation (relative to the transepicondylar axis) were determined. After fixation of the chamfer cutting guide to the resected surface of the distal femur, resection of the other four femoral surfaces (anterior cortex, posterior condyle, anterior chamfer, and posterior chamfer) was performed. Subsequently, femoral box cutting was performed. The proximal rod of the tibial extramedullary resection guide was placed in the center of the tibia. After inclining the cutting block by 3˚ by placing one finger on the tuberosity of the proximal tibia and two fingers on the ankle, tibial cutting was performed with a slope of 0˚–1˚ posteriorly in the sagittal plane. Subsequently, the femoral and tibial trial implants and the liner thickness were evaluated. The femoral and tibial implants were implanted using bone cement, and a polyethylene liner was fixed between them.

## Data analysis

All patients underwent follow-up radiography at 2 weeks, 6 weeks, 3 months, 6 months, 9 months, and 12 months after surgery, and annually thereafter. Clinical evaluations were performed using the KSS rating system [16]. The results were classified as excellent (80–100 points), good (70–79 points), fair (60–69 points), or poor (<60 points). Radiographs obtained after 6 weeks were used as a baseline for radiographic comparisons. Standing anteroposterior radiographs of both lower extremities were evaluated to assess the mechanical axis (hip–knee–ankle), coronal position of the femur and tibia (varus/valgus, α and β), and LLD. Lateral radiographs were obtained to assess the sagittal position of the tibia (posterior/anterior slope, δ). All radiographs were evaluated by two independent observers [18] (Fig 1). The polyethylene liner thickness was also examined. Outliers were defined as measured angles exceeding a 3˚ deviation from the neutral alignment and a liner thickness ≥16 mm on the radiographs. The mean error values and outliers for each study group were calculated and compared to determine the accuracy of the postoperative component positioning and the thickness of the polyethylene liner.

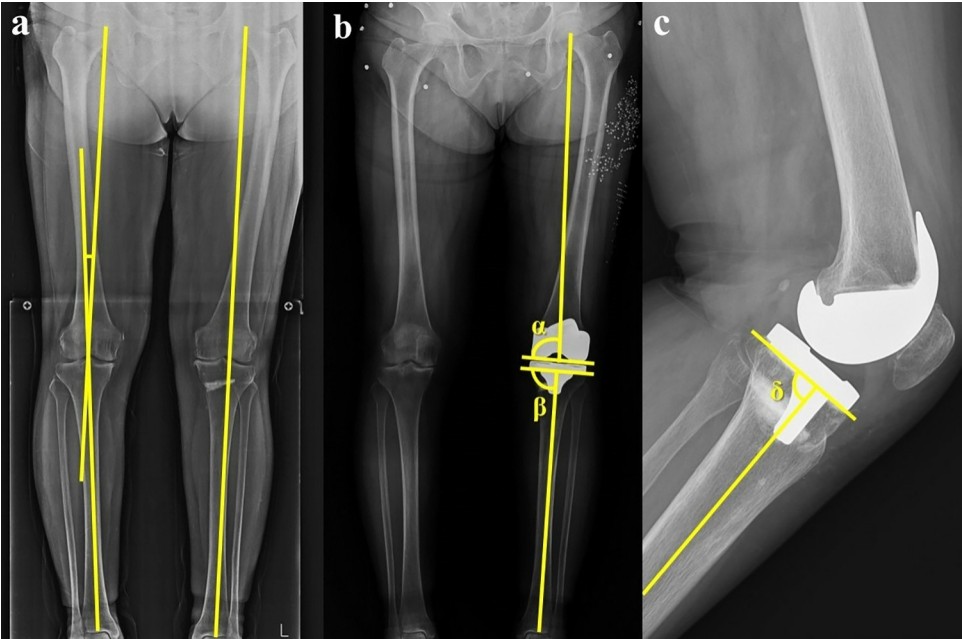

**Fig 1.** (**a**) The mechanical axis of the leg is the angle between the mechanical axis of the femur and tibia (Right leg). Leg length is measured from the center of the femoral head to the center of the talus and ankle (Left leg). (**b**) Radiologic measurement of the femoral and tibial implants. α, coronal inclination of the femoral component with the mechanical axis of the femur; β, coronal inclination of the tibial component with the mechanical axis of the tibia. (**c**) δ, sagittal inclination of the tibial component with the mechanical axis of the tibia.

## Statistical analysis

The sample size was determined using the G-power program (https://www.g-power.com/en/) [19]. For the sample size analysis, we considered an effect size of 0.5, an alpha significance level of 0.05, and a statistical power of 0.8. The appropriate sample size for the robotic group was determined to be 43 people. The sample size of the robotic group in this study was 30, which may lack statistical power. Therefore, we compared the two groups using propensity score matching to increase statistical power. Propensity scores were calculated for age, gender, and BMI, with robotic and conventional groups matched in a 1:2 ratio.

Student's t-test was used to analyze age, body mass index, and time from HTO to TKA. The chi-squared test was used to analyze sex, diagnosis, and KS knee and function scores. Data are presented as the mean ± standard deviation. Intraclass correlation coefficients (ICCs) were calculated using SPSS version 20 to determine the correlation between the measurements of the two independent observers. A common approach to quantifying the reliability of a measurement process is to calculate the ICC with a confidence interval. ICC is a statistical estimate that measures the degree of agreement between at least two quantitative measurement values. It is designed to measure the degree of reliability, consistency, stability, and agreement. For all analyses, a *p*-value of <0.05 indicated statistical significance.

## Results

The postoperative KSSs in the robotic and conventional groups were not statistically different at a mean follow-up of 2 years (Table 2). The mean KS knee scores in the robotic and conventional groups improved from 36.2 and 35.4 points preoperatively to 79.5 and 75.0 points at the final follow-up, respectively (*p* = 0.121). Mean preoperative function scores in the robotic and

**Table 2. Comparison of Knee Society score between the both groups.**

| | Robotic group | Conventional group | *p*-value |
|---|---|---|---|
| Postoperative scores | | | |
| Knee Society knee score (mean ± SD) | 79.5 ± 11.3 | 75.0 ± 13.6 | 0.121 |
| Excellent or good | 27 pts | 52 pts | |
| Fair | 2 pts | 6 pts | |
| Poor | 1 pt | 2 pts | |
| Function score (mean ± SD) | 79.8 ± 14.7 | 75.8 ± 13.3 | 0.197 |
| Excellent or good | 27 pts | 53 pts | |
| Fair | 2 pts | 5 pts | |
| Poor | 1 pt | 2 pts | |

SD: standard deviation, pts: patients

conventional groups improved from 38.7 and 38.2 points to 79.8 and 75.8 points at the final follow-up, respectively ($p = 0.197$).

The mean postoperative mechanical axis was 1.7˚ in the robotic group and 2.4˚ in the conventional group ($p < 0.05$). Postoperatively, the robotic group showed higher accuracy than the conventional group in terms of femur coronal (α), tibial coronal (β), and tibial sagittal inclination (δ) ($p < 0.05$). Additionally, LLD showed a smaller difference in the robotic group than in the conventional group ($p < 0.05$) (Table 3). A postoperative outlier in the mechanical

**Table 3. Comparison of radiologic results, LLD, and polyethylene liner thickness between the robotic and conventional groups.**

| | Robotic group ($n = 30$) | Conventional group ($n = 60$) | *p*-value |
|---|---|---|---|
| Preoperative (degree) | | | |
| HKA axis | 4.1 ± 3.2 | 4.4 ± 4.5 | n.s. |
| Posterior slope | 8.7 ± 3.9 | 9.0 ± 4.2 | n.s. |
| LLD (mm) | 6.2 ± 5.0 | 6.5 ± 4.3 | n.s. |
| Postoperative (degree) | | | |
| HKA axis | 1.7 ± 0.9 | 2.4 ± 1.6 | <0.05 |
| α | 90.0 ± 1.4 | 91.6 ± 1.8 | <0.05 |
| β | 90.3 ± 1.1 | 91.3 ± 1.5 | <0.05 |
| δ | 90.5 ± 2.3 | 91.4 ± 2.6 | <0.05 |
| LLD (mm) | 2.2 ± 2.4 | 7.0 ± 5.1 | <0.05 |
| Postoperative outliers, *n* (%) | | | |
| HKA axis | 4 (13.3) | 21 (35.0) | <0.05 |
| α | 1 (3.3) | 15 (25.0) | <0.05 |
| β | 0 (0) | 7 (11.7) | n.s. |
| δ | 3 (10.0) | 20 (33.3) | <0.05 |
| Polyethylene liner thickness, *n* (%) | | | <0.05 |
| 9 mm | 21 (70.0) | 17 (28.3) | |
| 11 mm | 7 (23.3) | 29 (48.3) | |
| 13 mm | 2 (6.7) | 10 (16.7) | |
| 16 mm | 0 (0) | 3 (5.0) | |
| 19 mm | 0 (0) | 1 (1.7) | |

*HKA*, hip–knee–ankle; *LLD*, leg length discrepancy; *α*, coronal inclination of the femoral component; *β*, coronal inclination of the tibial component; *δ*, sagittal inclination of the tibial component; *n.s.*, not significant.

**Table 4. Intraclass correlation coefficients between the two observers post surgery.**

| | Intraclass correlation | |
| | Robotic TKA | Conventional TKA |
|---|---|---|
| HKA axis | 0.997 (0.994–0.999) | 0.995 (0.992–0.997) |
| α | 0.908 (0.807–0.956) | 0.963 (0.938–0.978) |
| β | 0.960 (0.915–0.981) | 0.960 (0.933–0.976) |
| δ | 0.983 (0.963–0.992) | 0.970 (0.950–0.982) |
| LLD | 0.774 (0.525–0.892) | 0.844 (0.739–0.907) |

HKA, hip–knee–ankle; LLD, leg length discrepancy; α, coronal inclination of the femoral component; β, coronal inclination of the tibial component; δ, sagittal inclination of the tibial component.
The intraclass correlation coefficient between the two observers was within the 95% confidence interval. <0.5, poor; 0.50–0.75, moderate; 0.75–0.90, good; >0.90, excellent.

axis (>3˚) was observed in 4 patients (13.3%) in the robotic group and 21 patients (35.0%) in the conventional group, respectively ($p < 0.05$). A significant difference in the polyethylene liner thickness was observed between the two groups ($p < 0.05$) (Table 3). The intraclass correlation coefficient ranged from 0.75 to 0.99, within the 95% confidence interval, indicating a good correlation between the measurements of the two observers (Table 4).

## Discussion

Conversion of failed HTO to TKA is technically more challenging than primary TKA due to changes in mechanical axis and tibial anatomy, the complexity of the approach, and ligamentous imbalance. This retrospective case–control study aimed to compare the clinical outcomes, mechanical axis, component positioning, LLD, and polyethylene liner thickness between robotic-assisted TKA and conventional TKA in patients with failed HTO. The most significant finding of this study is that robotic-assisted TKA can improve the accuracy of mechanical axis alignment, component positioning, and polyethylene liner thickness and reduce LLD compared with conventional TKA in patients with failed HTO.

Performing conversion TKA after failed HTO requires careful surgical planning and attention because of the various anatomical deformities that occur following osteotomy [7, 20]. In this study, robotic surgery showed better accuracy than conventional surgery in terms of the implants needed to restore balance, alignment, and stability. In conventional surgery, especially in failed HTO, where the existing anatomical order was disrupted due to overcorrection or undercorrection, there were many difficulties in determining the amount of tibial cutting due to confusion of the femoral alignment guide and tibial valgus. However, in robotic surgery, the uncertainty of mechanical alignment restoration can be minimized based on preoperative CT images, and ligament balancing and accurate implant positioning can be achieved through real knee bone mapping. Additionally, although not statistically different, KS knee and function scores were higher in the robotic group at a mean follow-up of 2 years. Further long-term follow-up is likely required. According to Chen et al. [8], the conversion TKA group after HTO had significantly higher reoperation rates and complications than those in the primary TKA group. They explained that the joint line on the tibial side becomes valgus after osteotomy, and bone deficiencies on the tibial side can be confusing. Using conventional TKA procedures to determine femoral component rotation is often misleading and results in tibial misalignment. Parvizi et al. [7] reported lower KS pain scores in the converted TKA group after HTO with a higher incidence of aseptic loosening during a 15-year follow-up period. Postoperative limb misalignment and poor implant positioning are predictors of implant

failure following TKA. Accurate implant positioning and mechanical alignment can improve patient function and implant longevity. Therefore, difficulties, such as achieving accurate alignment and managing unintended anatomical changes after HTO, can be solved using robotic-assisted TKA. The application of robotic-assisted systems for conversion TKA is considered a reasonable technical option for managing failed HTO.

In this study, a significant difference in the polyethylene liner thickness was observed between the two groups ($p < 0.05$). Additionally, the results showed 0% (0/30) liner outliers in robotic-assisted TKA compared with 6.7% (4/60) in conventional TKA. A thicker liner is required to achieve ligament balance during TKA, which may be associated with intraoperative difficulties or errors [21]. Furthermore, poor implant survival rates have been reported in patients with thick liners [21]. The use of thicker polyethylene liners is associated with the elevation of the knee joint line and adversely affects the range of motion, mid-flexion stability, patellofemoral joint mechanics, and functional outcomes [22]. Robotic-assisted TKA after failed HTO may lead to consistent and accurate implant positioning and joint line restoration compared to conventional TKA, which may prolong implant longevity.

LLD after TKA is a common complaint that can diminish patient satisfaction [23]. Additionally, achieving proper limb alignment and length is critical for optimal function, stability, and patient satisfaction after TKA [24, 25]. However, inadequate LLD of the bilateral lower limbs after surgery can lead to complications such as gait abnormalities, increased joint stress, knee pain, discomfort, and increased risk of implant failure [26]. Residual LLD can occur when limb length is not properly accounted for during preoperative planning or addressed intraoperatively. However, robotic-assisted TKA has the advantage of sufficient preoperative planning with preoperative CT scans to minimize LLD. This study showed that LLD was less observed in robotic-assisted TKA than in conventional TKA. Proper restoration of the mechanical axis and joint space in robotic-assisted TKA can result in less LLD than that in conventional TKA.

To summarize, in patients with failed HTO with altered anatomy, robotic-assisted surgery has distinct advantages over conventional surgery. First, preoperative CT images determine surgical planning, including the implant size and position, leg alignment, LLD restoration, liner thickness, etc. Second, data are collected during surgery via real knee bone mapping; therefore, it is possible to combine them with preoperative CT images to determine the final implant position, size, and amount of bone cut without confounding the altered anatomy. Finally, there is an issue that is not specifically addressed in this paper. In cases where retained hardware is present in patients with failed HTO, it must be removed in conventional surgery before surgery can proceed. However, robotic-assisted surgery can be performed without removing the hardware, thereby saving time [27].

This study has several limitations. First, this was a single-center and retrospective study. Multicenter and prospective studies are needed to provide more robust and generalizable results. Second, a specific robotic system was used and compared with conventional joint surgery, which may limit the generalizability of the findings to all robotic systems. Third, plain radiographs were used instead of postoperative CT scans to measure the accuracy of component positioning and reduce the financial burden on patients. Further studies using postoperative CT scans are necessary to confirm and generalize these results because CT scans can provide more precise and detailed measurements.

## Conclusion

Conversion TKA after HTO failure requires careful surgical planning and attention due to the various anatomic variations that occur after osteotomy. This study showed that robotic-

assisted TKA could improve the mechanical axis, increase the accuracy of component positioning and polyethylene liner thickness, and reduce LLD compared to those in conventional TKA patients with failed HTO. The application of a robotic-assisted system used in conversion TKA is considered a reasonable technical option for managing failed HTO. Further multicenter studies with long-term follow-up measures are necessary to determine whether the accuracy of robotic-assisted TKA can translate into better clinical outcomes and patient satisfaction.

## Author Contributions

**Conceptualization:** Ji-Hoon Baek, Chang Hyun Nam.

**Investigation:** Taehyeon Kim, Hye Sun Ahn.

**Methodology:** Dong Nyoung Lee, Juneyoung Heo.

**Project administration:** Ji-Hoon Baek.

**Resources:** Su Chan Lee.

**Supervision:** Su Chan Lee.

**Validation:** Ji-Hoon Baek.

**Writing – original draft:** Ji-Hoon Baek.

**Writing – review & editing:** Ji-Hoon Baek, Chang Hyun Nam.

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
