## [Decision Letter · Decision Letter 0]

20 Aug 2024

PONE-D-24-23964Better accuracy of robotic-assisted total knee arthroplasty compared to conventional technique in patients with failed high tibial osteotomyPLOS ONE

Dear Dr. Nam,

Thank you for submitting your manuscript to PLOS ONE. After careful consideration, we feel that it has merit but does not fully meet PLOS ONE’s publication criteria as it currently stands. Therefore, we invite you to submit a revised version of the manuscript that addresses the points raised during the review process.

The reviewers highlighted some critical issues in the article you sent us. Please read the reviewers' comments carefully. A major revision is required in order for this article to be published.

We look forward to receiving your revised manuscript.

Kind regards,

Gennaro Pipino, Md

Academic Editor

PLOS ONE

5. We note that Figure 1 in your submission contain copyrighted images. All PLOS content is published under the Creative Commons Attribution License (CC BY 4.0), which means that the manuscript, images, and Supporting Information files will be freely available online, and any third party is permitted to access, download, copy, distribute, and use these materials in any way, even commercially, with proper attribution. For more information, see our copyright guidelines: http://journals.plos.org/plosone/s/licenses-and-copyright.

Reviewers' comments:

Reviewer's Responses to Questions

**Comments to the Author**

1. Is the manuscript technically sound, and do the data support the conclusions?

Reviewer #1: Yes

Reviewer #2: Yes

Reviewer #3: Partly

2. Has the statistical analysis been performed appropriately and rigorously? 

Reviewer #1: Yes

Reviewer #2: No

Reviewer #3: Yes

3. Have the authors made all data underlying the findings in their manuscript fully available?

Reviewer #1: Yes

Reviewer #2: Yes

Reviewer #3: Yes

4. Is the manuscript presented in an intelligible fashion and written in standard English?

Reviewer #1: Yes

Reviewer #2: Yes

Reviewer #3: No

5. Review Comments to the Author

Reviewer #1: 

Dear Authors, I had the opportunity to read your article, which has a clearly defined objective of comparing robot-assisted TKA with conventional TKA in patients with failed HTO. The methodological structure of the study is well articulated, with details on patient selection, inclusion criteria, and procedures performed. The results are clearly presented, with statistically significant differences well highlighted. The detailed description of the surgical procedures for both techniques (robotic and conventional) is useful for understanding the operational differences. This implies that the physician must necessarily provide appropriate information to the patient about the potential risks, benefits, and specific advantages of one procedure versus the other. In this context, I would suggest an important work that I recommend citing in your references (https://doi.org/10.1016/j.jflm.2024.102674), which clearly states that in conducting a thorough informed consent process, physicians have the professional duty to present and initiate a discussion on the risks, benefits, and possible alternatives to a given procedure. Illustrating and discussing possible alternatives is a fundamental element of the disclosure process; patients may not be able to assess risks in abstract terms and would therefore need to rely on a comparative framework to make a truly informed decision. Addressing these discussion elements in detail would certainly contribute to making the article more comprehensive, which is advisable, given that the article, while presenting quality and strengths, needs appropriate elaborations. The inclusion of details such as the measurement of the mechanical axis, alignment, and ligament balancing provides an in-depth view of the methods used. The results showing greater accuracy in mechanical axis and component positioning in the robotic group are well supported by statistical data. The discussion on the importance of alignment and ligament balancing for implant longevity and patient satisfaction is well-argued. In light of these elements—very representative and entirely agreeable—there are some areas that need appropriate elaborations. In this regard, although the study used a specific robotic system, no detailed information is provided on which system was used. It would be useful to know the type of robotic system employed to better understand the results and their generalizability. The lack of clinical follow-up data is a significant limitation. Additional information on long-term clinical outcomes could strengthen the conclusions. The fact that the study is single-center and retrospective is another limitation. Multicenter and prospective studies could provide more robust and generalizable results. The use of simple radiographs instead of postoperative CT scans to measure the accuracy of component positioning is a methodological choice that could be discussed more thoroughly. CT scans provide more precise and detailed measurements, although they may increase costs for patients. The mention of intraclass correlation coefficients between the measurements of the two observers is positive, but details on how these coefficients were calculated and interpreted are lacking. More details on these statistical aspects could better clarify the validity of the measurements. It is evident that the surgeon's competence and experience in performing the procedures can significantly influence the results. In conclusion, the article provides a useful comparison between robot-assisted TKA and conventional TKA in patients with failed HTO, highlighting the benefits of robotic assistance in terms of precision and alignment. However, further studies with long-term clinical follow-ups, use of postoperative CT scans, and involvement of multiple centers are necessary to confirm and generalize these results. Elaboration on methodological and statistical details would further strengthen the study.

Reviewer #2: 

First of all, congratulations on the good results of TKR in HTO failure patients using robots.

Let me point out a few things.

1. Please describe exactly what the definition of failure in HTO

2. It is unclear whether the advantages of robot surgery in the Ligament balance and the advantages of robot surgery in the valgus state of the tibia due to HTO are advantageous in holding the guide over conventional or in the positioning of the tibial component. Describe the advantages of an accurate robot in this regard

3. This result does not include clinical results but only radiologic results, such as mechanical axis and component slope. The discussion mentions the long-term follow-up results, but this does not appear appropriate. Please correct it.

4. The analysis of the advantages of robotic surgery in the discussion is not clear. Describe exactly and accureately the advantages of robot-assisted surgery over conventional techniques.

Reviewer #3: 

We would like to thank the authors for attempting to evaluate robotic technique in TKA, especially after conversion due to HTO.

However, the adequacy of statistical analysis based on tele-radiographs, which do not take into account the possibility of rotational imaging (the gold standard - multispiral computed tomography) due to the human factor and leading to conclusions on the scale of a degree, raises great doubts about their correctness.

At the same time, the simple assessment of the mechanical axis deviation does not give any insight into the clinical-functional results, which were not evaluated using appropriate scales (example KSS, WOMAC, etc.), which casts doubt on the scientific value of the work, while the number of patients in a methodologically correct approach would allow valuable conclusions to be drawn.

The "Aim of the study" is missing in the materials and methods of the main manuscript. The "Discussion" block comes after the "Results", but there are no "Conclusions", which is probably an editing error. The English language needs to be double-checked by a native speaker.

The reference list needs to be supplemented with current sources, 14 out of 23 references are older than 10 years, including publications that have only museum value in 2024...

6. PLOS authors have the option to publish the peer review history of their article (what does this mean?). If published, this will include your full peer review and any attached files.

Reviewer #1: **Yes: **Giuseppe Basile

Reviewer #2: No

Reviewer #3: No

---

## [Author Response · Author response to Decision Letter 0]

1 Oct 2024

Response to reviewers

Reviewer #1: 

Dear Authors, I had the opportunity to read your article, which has a clearly defined objective of comparing robot-assisted TKA with conventional TKA in patients with failed HTO. The methodological structure of the study is well articulated, with details on patient selection, inclusion criteria, and procedures performed. The results are clearly presented, with statistically significant differences well highlighted. The detailed description of the surgical procedures for both techniques (robotic and conventional) is useful for understanding the operational differences. 

This implies that the physician must necessarily provide appropriate information to the patient about the potential risks, benefits, and specific advantages of one procedure versus the other. In this context, I would suggest an important work that I recommend citing in your references (https://doi.org/10.1016/j.jflm.2024.102674), which clearly states that in conducting a thorough informed consent process, physicians have the professional duty to present and initiate a discussion on the risks, benefits, and possible alternatives to a given procedure. Illustrating and discussing possible alternatives is a fundamental element of the disclosure process; patients may not be able to assess risks in abstract terms and would therefore need to rely on a comparative framework to make a truly informed decision. Addressing these discussion elements in detail would certainly contribute to making the article more comprehensive, which is advisable, given that the article, while presenting quality and strengths, needs appropriate elaborations.

Response: We have added “During this study, we provided sufficient and appropriate information about the respective potential risks, benefits, and specific advantages of robotic and conventional surgeries [15].” in the Materials and Methods.

Response: We have added “15. Bolcato V, Franzetti C, Fassina G, Basile G, Martinez RM, Tronconi LP. Comparative study on informed consent regulation in health care among Italy, France, United Kingdom, Nordic Countries, Germany, and Spain. J Forensic Leg Med. 2024;103:102674. doi:10.1016/j.jflm.2024.102674” in the References.

The inclusion of details such as the measurement of the mechanical axis, alignment, and ligament balancing provides an in-depth view of the methods used. The results showing greater accuracy in mechanical axis and component positioning in the robotic group are well supported by statistical data. The discussion on the importance of alignment and ligament balancing for implant longevity and patient satisfaction is well-argued. In light of these elements—very representative and entirely agreeable—there are some areas that need appropriate elaborations. 

In this regard, although the study used a specific robotic system, no detailed information is provided on which system was used. It would be useful to know the type of robotic system employed to better understand the results and their generalizability.

Response: We have added “The robotic-assisted TKA system (Mako) was introduced to our hospital in 2020. Mako (Stryker, Kalamazoo, MI, USA) is a leading semi-active robotic system and is currently the most commonly used surgical robot for joint operations worldwide [11]. Mako uses preoperative lower extremity computed tomography (CT) images to preplan the size and position of the implant. During surgery, actual knee bone information and preoperative CT data are used to determine the implant size, position, and extent of bone resection. The bone cutting is performed with a cutting saw mounted on the robotic arm. This robotic arm provides haptic feedback and stops the saw when it starts to exceed a preset range while cutting the bone, preventing damage to soft tissue.” in the Materials and Methods.

Response: We have added “11. Ma N, Sun P, Xin P, Zhong S, Xie J, Xiao L. Comparison of the efficacy and safety of MAKO robot-assisted total knee arthroplasty versus conventional manual total knee arthroplasty in uncomplicated unilateral total knee arthroplasty a single-centre retrospective analysis. Int Orthop. 2024;48(9):2351-2358. doi:10.1007/s00264-024-06234-0” in the References.

The lack of clinical follow-up data is a significant limitation. Additional information on long-term clinical outcomes could strengthen the conclusions.

Response: We have added “The postoperative KSSs in the robotic and conventional groups were not statistically different at a mean follow-up of 2 years (Table 2). The mean KS knee scores in the robotic and conventional groups improved from 36.2 and 35.4 points preoperatively to 79.5 and 75.0 points at the final follow-up, respectively (p = 0.121). Mean preoperative function scores in the robotic and conventional groups improved from 38.7 and 38.2 points to 79.8 and 75.8 points at the final follow-up, respectively (p = 0.197).” in the Results.

Response: We have added “Table 2” in the Results.

The fact that the study is single-center and retrospective is another limitation. Multicenter and prospective studies could provide more robust and generalizable results.

Response: We have changed from “This study has several limitations. First, this was a retrospective study and did not include clinical follow-up results. Second, this was a single-center study, and a specific robotic system was used and compared with conventional joint surgery, which may limit the generalizability of the findings to all robotic systems. Finally, plain radiographs were used instead of postoperative CT scans to measure the accuracy of component positioning to reduce the financial burden on patients.” to “This study has several limitations. First, this was a single-center and retrospective study. Multicenter and prospective studies are needed to provide more robust and generalizable results. Second, a specific robotic system was used and compared with conventional joint surgery, which may limit the generalizability of the findings to all robotic systems. Third, plain radiographs were used instead of postoperative CT scans to measure the accuracy of component positioning and reduce the financial burden on patients. Further studies using postoperative CT scans are necessary to confirm and generalize these results because CT scans can provide more precise and detailed measurements.” in the Limitation part.

The use of simple radiographs instead of postoperative CT scans to measure the accuracy of component positioning is a methodological choice that could be discussed more thoroughly. CT scans provide more precise and detailed measurements, although they may increase costs for patients.

Response: We have changed from “This study has several limitations. First, this was a retrospective study and did not include clinical follow-up results. Second, this was a single-center study, and a specific robotic system was used and compared with conventional joint surgery, which may limit the generalizability of the findings to all robotic systems. Finally, plain radiographs were used instead of postoperative CT scans to measure the accuracy of component positioning to reduce the financial burden on patients.” to “This study has several limitations. First, this was a single-center and retrospective study. Multicenter and prospective studies are needed to provide more robust and generalizable results. Second, a specific robotic system was used and compared with conventional joint surgery, which may limit the generalizability of the findings to all robotic systems. Third, plain radiographs were used instead of postoperative CT scans to measure the accuracy of component positioning and reduce the financial burden on patients. Further studies using postoperative CT scans are necessary to confirm and generalize these results because CT scans can provide more precise and detailed measurements.” in the Limitation part.

The mention of intraclass correlation coefficients between the measurements of the two observers is positive, but details on how these coefficients were calculated and interpreted are lacking. More details on these statistical aspects could better clarify the validity of the measurements.

Response: We have changed from “Differences in variables between the two groups were evaluated using the Mann–Whitney U test. Data are presented as the mean ± standard deviation. Intraclass correlation coefficients were calculated using SPSS version 20 to determine the correlation between the measurements of the two independent observers. For all analyses, a p-value < 0.05 indicated statistical significance.” to “Student’s t-test was used to analyze age, body mass index, and time from HTO to TKA. The chi-squared test was used to analyze sex, diagnosis, and KS knee and function scores. Data are presented as the mean ± standard deviation. Intraclass correlation coefficients (ICCs) were calculated using SPSS version 20 to determine the correlation between the measurements of the two independent observers. A common approach to quantifying the reliability of a measurement process is to calculate the ICC with a confidence interval. ICC is a statistical estimate that measures the degree of agreement between at least two quantitative measurement values. It is designed to measure the degree of reliability, consistency, stability, and agreement. For all analyses, a p-value of <0.05 indicated statistical significance.” in the Statistical analysis part.

It is evident that the surgeon's competence and experience in performing the procedures can significantly influence the results. In conclusion, the article provides a useful comparison between robot-assisted TKA and conventional TKA in patients with failed HTO, highlighting the benefits of robotic assistance in terms of precision and alignment. 

However, further studies with long-term clinical follow-ups, use of postoperative CT scans, and involvement of multiple centers are necessary to confirm and generalize these results. Elaboration on methodological and statistical details would further strengthen the study.

Response: We have changed from “In conclusion, this study showed that robotic-assisted TKA can improve the mechanical axis, increase the accuracy of component positioning and polyethylene liner thickness, and reduce LLD compared with those in conventional TKA in patients with failed HTO. Further larger-scale studies with long-term follow-up measures are necessary to determine whether the accuracy of robotic-assisted TKA can translate into better clinical outcomes and patient satisfaction.” to “Conversion TKA after HTO failure requires careful surgical planning and attention due to the various anatomic variations that occur after osteotomy. This study showed that robotic-assisted TKA could improve the mechanical axis, increase the accuracy of component positioning and polyethylene liner thickness, and reduce LLD compared to those in conventional TKA patients with failed HTO. The application of a robotic-assisted system used in conversion TKA is considered a reasonable technical option for managing failed HTO. Further multicenter studies with long-term follow-up measures are necessary to determine whether the accuracy of robotic-assisted TKA can translate into better clinical outcomes and patient satisfaction.” in the Conclusion part.

Reviewer #2: 

First of all, congratulations on the good results of TKR in HTO failure patients using robots.

Let me point out a few things.

1. Please describe exactly what the definition of failure in HTO.

Response: We have added “Failure of HTO is defined as the need for conversion to total knee arthroplasty (TKA) since HTO is offered as an option to delay the need for knee arthroplasty.” in the Introduction.

2. It is unclear whether the advantages of robot surgery in the Ligament balance and the advantages of robot surgery in the valgus state of the tibia due to HTO are advantageous in holding the guide over conventional or in the positioning of the tibial component. Describe the advantages of an accurate robot in this regard.

Response: We have changed from “Performing conversion TKA after failed HTO requires careful surgical planning and attention because of the various anatomical deformities that occur following osteotomy [7,16]. According to Chen et al. [8], the conversion TKA group after HTO had significantly higher reoperation rates and complications than those in the primary TKA group. They explained that the joint line on the tibial side becomes valgus after osteotomy, and bone deficiencies on the tibial side can be confusing. Using conventional TKA procedures to determine femoral component rotation is often misleading and results in tibial misalignment. Parvizi et al. [7] reported lower Knee Society pain scores in the converted TKA group after HTO with a higher incidence of aseptic loosening during a 15-year follow-up period. Postoperative limb misalignment and poor implant positioning are predictors of implant failure following TKA. Accurate implant positioning and mechanical alignment can improve patient function and implant longevity. Therefore, difficulties, such as achieving accurate alignment and managing unintended anatomical changes after HTO, can be solved using robotic-assisted TKA.” to “Performing conversion TKA after failed HTO requires careful surgical planning and attention because of the various anatomical deformities that occur following osteotomy [7,20]. In this study, robotic surgery showed better accuracy than conventional surgery in terms of the implants needed to restore balance, alignment, and stability. In conventional surgery, especially in failed HTO, where the existing anatomical order was disrupted due to overcorrection or undercorrection, there were many difficulties in determining the amount of tibial cutting due to confusion of the femoral alignment guide and tibial valgus. However, in robotic surgery, the uncertainty of mechanical alignment restoration can be minimized based on preoperative CT images, and ligament balancing and accurate implant positioning can be achieved through real knee bone mapping. Additionally, although not statistically different, KS knee and function scores were higher in the robotic group at a mean follow-up of 2 years. Further long-term follow-up is likely required. According to Chen et al. [8], the conversion TKA group after HTO had significantly higher reoperation rates and complications than those in the primary TKA group. They explained that the joint line on the tibial side becomes valgus after osteotomy, and bone deficiencies on the tibial side can be confusing. Using conventional TKA procedures to determine femoral component rotation is often misleading and results in tibial misalignment. Parvizi et al. [7] reported lower KS pain scores in the converted TKA group after HTO with a higher incidence of aseptic loosening during a 15-year follow-up period. Postoperative limb misalignment and poor implant positioning are predictors of implant failure following TKA. Accurate implant positioning and mechanical alignment can improve patient function and implant longevity. Therefore, difficulties, such as achieving accurate alignment and managing unintended anatomical changes after HTO, can be solved using robotic-assisted TKA. The application of robotic-assisted systems for conversion TKA is considered a reasonable technical option for managing failed HTO.” in the Discussion.

3. This result does not include clinical results but only radiologic results, such as mechanical axis and component slope. The discussion mentions the long-term follow-up results, but this does not appear appropriate. Please correct it.

Response: We have added “The postoperative KSSs in the robotic and conventional groups were not statistically different at a mean follow-up of 2 years (Table 2). The mean KS knee scores in the robotic and conventional groups improved from 36.2 and 35.4 points preoperatively to 79.5 and 75.0 points at the final follow-up, respectively (p = 0.121). Mean preoperative function scores in the robotic and conventional groups improved from 38.7 and 38.2 points to 79.8 and 75.8 points at the final follow-up, respectively (p = 0.197).” in the Results.

Response: We have added “Table 2” in the Results.

4. The analysis of the advantages of roboti

---

## [Decision Letter · Decision Letter 1]

23 Oct 2024

Better accuracy of robotic-assisted total knee arthroplasty compared to conventional technique in patients with failed high tibial osteotomy

PONE-D-24-23964R1

Dear Dr. Nam,

We’re pleased to inform you that your manuscript has been judged scientifically suitable for publication and will be formally accepted for publication once it meets all outstanding technical requirements.

Kind regards,

Gennaro Pipino, Md

Academic Editor

PLOS ONE

Additional Editor Comments (optional):

Thanks to the authors for sending us this paper. All the reviewers interviewed gave unanimous feedback on the quality of your work. Congratulations on the work done. It is ready for publication

Reviewers' comments:

Reviewer's Responses to Questions

**Comments to the Author**

1. If the authors have adequately addressed your comments raised in a previous round of review and you feel that this manuscript is now acceptable for publication, you may indicate that here to bypass the “Comments to the Author” section, enter your conflict of interest statement in the “Confidential to Editor” section, and submit your "Accept" recommendation.

Reviewer #1: All comments have been addressed

Reviewer #3: All comments have been addressed

2. Is the manuscript technically sound, and do the data support the conclusions?

Reviewer #1: Yes

Reviewer #3: Yes

3. Has the statistical analysis been performed appropriately and rigorously? 

Reviewer #1: Yes

Reviewer #3: Yes

4. Have the authors made all data underlying the findings in their manuscript fully available?

Reviewer #1: Yes

Reviewer #3: Yes

5. Is the manuscript presented in an intelligible fashion and written in standard English?

Reviewer #1: Yes

Reviewer #3: Yes

6. Review Comments to the Author

Reviewer #1: Dear Authors, I have read your additions, finding them entirely consistent and in accordance with my expectations. I consider your work interesting. The purpose is clear and respected. I believe that the information provided is to be considered entirely sufficient and represents useful elements to encourage the development of new scientific work.

Reviewer #3: Thanks a lot to the team of authors for the work done and increasing the value of the findings of the paper materials.

7. PLOS authors have the option to publish the peer review history of their article (what does this mean?). If published, this will include your full peer review and any attached files.

Reviewer #1: **Yes: **Giuseppe Basile

Reviewer #3: No

---

## [Editor Report · Acceptance letter]

29 Oct 2024

PONE-D-24-23964R1 

PLOS ONE

Dear Dr. Nam, 

I'm pleased to inform you that your manuscript has been deemed suitable for publication in PLOS ONE. Congratulations! Your manuscript is now being handed over to our production team.

Kind regards, 

on behalf of

Professor Gennaro Pipino 

Academic Editor

PLOS ONE